# First crAss-Like Phage Genome Encoding the Diversity-Generating Retroelement (DGR)

**DOI:** 10.3390/v12050573

**Published:** 2020-05-22

**Authors:** Vera Morozova, Mikhail Fofanov, Nina Tikunova, Igor Babkin, Vitaliy V. Morozov, Artem Tikunov

**Affiliations:** Institute of Chemical Biology and Fundamental Medicine SB RAS, 630090 Novosibirsk, Russia; mvfofanov@mail.ru (M.F.); i_babkin@mail.ru (I.B.); doctor.morozov@mail.ru (V.V.M.); arttik@ngs.ru (A.T.)

**Keywords:** crAssphage, genome, diversity-generating retroelement (DGR), variable repeat, template repeat, initiation of mutagenic homing

## Abstract

A new crAss-like genome encoding diversity-generating retroelement (DGR) was found in the fecal virome of a healthy volunteer. The genome of the phage referred to as the crAssphage LMMB, belonged to the candidate *genus I* of the AlphacrAssvirinae subfamily. The DGR-cassette of the crAssphage LMMB contained all the essential elements: the gene encoding reverse transcriptase (RT), the target gene (TG) encoding the tail-collar fiber protein, and variable and template repeats (VR and TR) with IMH (initiation of mutagenic homing) and IMH* sequences at the 3′-end of the VR and TR, respectively. Architecture of the DGR-cassette was TG-VR(IMH)-TR(IMH*)-RT and an accessory variable determinant (avd) was absent from the cassette. Analysis of 91 genomes and genome fragments from *genus I* of the AlphacrAssvirinae showed that 15 (16%) of the genomes had DGRs with the same architecture as the crAssphage LMMB, while 66 of the genomes contained incomplete DGR-cassettes or some elements of the DGR.

## 1. Introduction

CrAss-like phages have been discovered by computational analysis of human fecal metagenome data and their genomes have been shown to be the most abundant group of sequences (up to 90%) in the human gut virome [1]. Subsequently, they were classified as the crAss-like group [2], and then it was proposed to allocate them into a separate family, consisting of four subfamilies and ten genera, namely AlphacrAssvirinae (genera *I, III, IV* and *IX*), BetacrAssvirinae (*VI*), GammacrAssvirinae (*II, V*) and DeltacrAssvirinae (*VII, VIII, X*) [3]. CrAss-like phages were predicted to infect bacteria of the phylum Bacteroidetes, which are the most widely represented bacteria in the intestinal tract of humans [1]. Bacteroidetes are difficult to cultivate and thus limited data are available about these viruses; however, the CrAss001 phage has been isolated and confirmed to have podoviral morphology and infect *Bacteroides intestinalis* [4].

Intensive studies on the biology, taxonomy, and role of crAss-like phages have shown that they have high levels of genetic diversity. Members of this group of phages have been found in a range of environments, including the human gut and feces, termite gut, terrestrial/groundwater environments, soda lakes (hypersaline brine), marine sediments, and plant roots [2,3,5,6,7]. CrAss-like phages likely infect a variety of bacterial hosts; however, the mechanisms generating this variability are unknown.

One of the remarkable systems responsible for the variability of prokaryotic microorganisms are the diversity generating retroelements (DGRs), which use reverse transcription to introduce huge numbers of nucleotide substitutions in specific target genes [8,9]. The DGR has been initially discovered in the genome of the temperate *Bordetella* phage BPP-1 [10], and found to provide changes in the host-recognizing structures of *Bordetella* phages, hence enabling phage adaptation to dynamic changes on the surface of the *Bordetella* host [10,11]. Subsequent genetic and metagenomics studies have shown that DGRs contain several essential elements. A crucial element of each DGR is the gene encoding reverse transcriptase (RT). This enzyme plays an important role in exchanging between two repeats, which have similar nucleotide sequences, during a process called mutagenic retrohoming [8,9,12]. One repeat is a template repeat (TR) while the other is a variable repeat (VR), the latter of which is often located at the 3′-end of the target gene. During mutagenic retrohoming, an RNA-transcript from the TR is reverse transcribed by the RT and almost all the adenines in the cDNA sequence could be subjected to A-to-N mutagenesis. This leads to changes in the VR sequence and, hence, in the corresponding *C*-terminal amino acid (aa) positions in the protein encoded by the target gene [9]. The initiation of the mutagenic homing (IMH) sequence is located at the 3′-end of the VR, while a non-identical IMH* repeat is usually found at the 3′-end of the TR. Additionally, the accessory variability determinant (avd) gene or, sometimes, a gene encoding a component of the bacterial efflux pump may be part of the DGR-cassette. The target gene, RT, VR, TR, IMH, and IMH* are essential elements of the DGR-cassette, while approximately one quarter of DGRs have no homologs to the accessory gene [8,9]. DGRs have been classified based on their architectural variations and the phylogeny of their specific elements [9].

DGRs have been found in the genomes of a wide range of microorganisms belonging to the Bacteroidetes, Cyanobacteria, Firmicutes, Proteobacteria, and Archaea phyla [13,14,15,16,17,18]. Most DGRs are associated with bacterial chromosomes, including the genomes of putative prophages [8,9,14]. Only a few DGRs containing the RT gene have been found in the genomes of free phages including the first finding in the *Bordetella* phage BPP-1 [10,12,19,20]. Here we describe the first complete genome of a crAss-like phage containing the DGR-cassette with the RT and other essential elements, and also analyze the occurrence of such cassettes in other relative crAss-like phages.

## 2. Materials and Methods

### 2.1. Viral DNA Isolation and Sequencing

A fecal sample (0.3 g) from a healthy volunteer was re-suspended in 1.2 mL of sterile phosphate-buffered saline (PBS, pH 7.5) and clarified by centrifugation 4 times at 20,000× *g* for 5 min at 4 °C. After every centrifugation, supernatant was transferred to a new sterile centrifugation tube. The final supernatant was treated with 5 U of DNase I (Thermo Fisher Scientific, Waltham, MA, USA) followed by treatment with 100 μg/mL of Proteinase K (Thermo Fisher Scientific) supplemented with EDTA and SDS to final concentrations of 20 mM and 0.5%, respectively. The mixture was incubated for 3 h at 55 °C. DNA was extracted from the whole volume of supernatant using a phenol-chloroform method with subsequent ethanol precipitation. Purified DNA was diluted in 50 μL of 0.1 × TE-buffer. The DNA was further used to construct a virome shotgun library using the NEB Next Ultra DNA library prep kit (New England Biolabs, Ipswich, MA, USA). Sequencing was carried out using a MiSeq Benchtop Sequencer (Illumina Inc., Foster City, CA, USA) in the SB RAS Genomics Core Facility, ICBFM SB RAS, Novosibirsk, Russia, and a MiSeq Reagent Kit 2 × 250 v.2 (Illumina Inc.). The obtained sequences were assembled *de novo* using the CLC Genomics Workbench software v.6.0. This study was approved by a local Ethics committee of the Center for Personalized Medicine in Novosibirsk; protocol #2, 12.02.2019.

### 2.2. Genome Analysis

The most abundant contig was compared with sequences deposited in the GenBank database using the BLASTN algorithm. The putative open reading frames (ORFs) were revealed using Vector NTI [21] and their functions were predicted by the BLASTX algorithm according to their similarity with annotated protein sequences from the GenBank database. CrAssphage genomes were downloaded from GenBank using the crAssphage LMMB nucleotide sequences of genes encoding the major capsid protein and RT as a query in a BLASTN search against nucleotide collection (nr/nt) and whole-genome shotgun contig (WGS) databases. The DGR sequences were identified manually using multiple alignments of putative DGR-containing genome sequences and, additionally, were verified using myDGR software [22]. MAFFT software (https://mafft.cbrc.jp/alignment/server) was used to align genome sequences and make a dot-plot analysis of the genomes. Alignment of RT sequences was performed using algorithm M-Coffee from the T-Coffee software package [23], and MEGA X software [24] was used for phylogenetic analysis of aligned sequences. A phylogenetic analysis of crAss-like phages was performed using the Viral Proteomic Tree server (ViPTree) [25]. Sequence logos for VRs were generated using WebLogo [26]. The CGView server was used for comparative analysis of the genome of crAssphage LMMB and the genomes of related phages [27]. The investigated nucleotide sequence was deposited to the GenBank database under accession number [MT006214].

## 3. Results

### 3.1. Analysis of the crAssphage LMMB Genome

Following the *de novo* assembly of the fecal virome of a healthy volunteer, several contigs with a length >40,000 bp were identified. One of the contigs had a length of 98,458 bp with an average coverage of 185, and this contig was found to have high similarity (identity level ~97% with the query cover of 95%) to the genome of the prototype p-crAssphage [NC_024711.1], which has been previously analyzed in detail [1,2]. The sequence similarity analysis performed using the CGView server revealed that the studied genome of a phage, referred to as the crAssphage LMMB, showed gene synteny typical of other crAss-like genomes (Figure 1) from the candidate *genus I* of the previously suggested subfamily AlphacrAssvirinae, which contains p-crAssphage [3].

The genome of the crAssphage LMMB comprised 81 putative ORFs with 38 ORFs located on the forward strand and 43 ORFs found on the reverse strand (Figure 1). Previously it was reported that all of the crAss-like phages of candidate genera *I, II*, and *IV* do not contain tRNA genes [3]. Among the annotated ORFs, no tRNA genes were found in the genome of the crAssphage LMMB.

The taxonomy of the crAssphage LMMB was confirmed using phylogenetic analysis of the putative proteomes of related crAss-like phages (Figure 2), and the phage was identified as a member of *genus I* of the subfamily AlphacrAssvirinae.

### 3.2. Identification and Characterization of the DGR in the Genome of crAssphage LMMB

A peculiarity of the crAssphage LMMB genome, which distinguished it from the reference genome of p-crAssphage [NC_024711.1], was the presence of the gene encoding the putative RT (protein ID QIN93296). This gene (ORF49) was located at the 3′-end of a cluster of structural genes, downstream the gene (ORF50), which encodes the tail-collar fiber protein (T-CFP, protein ID QIN93297) (Figure 1 and Figure 3). Detailed characterization of the genome regions upstream and downstream the RT showed that a cassette of essential DGR elements was located upstream the RT (Figure 3). This DGR-cassette contained the target gene encoding T-CFP, the VR at the 3′-end of the target gene, TR, and RT. Both the VR and TR had identical IMH and IMH* at their 3′-ends, respectively (Figure 3).

To clarify the type and taxonomy of the identified DGR, phylogenetic analysis of the RT from the crAssphage LMMB versus RT sequences extracted from the study of Wu et al. [9] was performed, and this RT was subsequently classified as a member of the lineage 3 (Figure 4). The major architecture of the core elements in DGRs with RT from lineage 3 is known to be “target gene-TR-RT” [9], which corresponded to the DGR-cassette found in the genome of the crAssphage LMMB. No avd gene sequence was revealed in the crAssphage LMMB DGR, which is similar to other DGRs from lineage 3. It has been shown that all the revealed VRs, which most directly correlated with RTs of lineage 3, corresponded to C-type lectin folds of major class 3 (CLec3) [9].

In the crAssphage LMMB, the target protein (T-CFP) consists of 486 aa, 35 of which can be defined as the VR. There are 35 codons in the TR (Figure 3), containing 28 adenines that can theoretically generate ~10^17 (428) DNA sequences in the VR that correspond to 10^16 aa sequences in the target protein. When the VR and TR were compared, 21 adenines in the structure of the VR were found to undergo A-to-N mutagenesis. Additionally, two non A-to-N mutations were detected in the VR (Figure 3).

### 3.3. Comparative Analysis of Putative DGRs in the Genomes of crAss-Like Phages from Genus I of the AlphacrAssvirinae

To investigate the prevalence of DGRs in crAss-like phages from the candidate *genus I* of the subfamily AlphacrAssvirinae, a total of 90 genomes were selected. The selected genomes included 63 genomes of *genus I* that were downloaded from Supplementary data [3], and 27 crAss-like genomes that were extracted from the GenBank database using the sequences of genes encoding major capsid protein and RT of the crAssphage LMMB as queries in a BLASTN search against the human WGS database. Dot-plot analysis of the genomes and phylogeny of the major capsid proteins confirmed that they belonged to the candidate *genus I* of the subfamily AlphacrAssvirinae (Appendix A).

To identify the presence of putative DGRs, 90 genomes were screened for the presence of the gene encoding T-CFP using BLASTN; however, it was found that only 58 genomes contained the gene and 22 genomes contained its fragment. Then, the ORF encoding the tail needle protein (protein ID QIN93298) was found in 88 screened genomes (two genomes did not contain the ORF), and genomic regions of approximately 7500 bp adjacent to this ORF were then compared. The analysis showed substantial variability in these investigated regions (Figure 5, Appendix A). DGRs containing all the essential elements of the retrohoming system (target gene with VR, TR, and RT) were found in 16% (15/91, including crAssphage LMMB) of the examined genomes (Figure 5, Appendix A). Architectures of these DGRs were identical and corresponded to the DGR-cassette in the genome of the crAssphage LMMB (Figure 5). The presence of core elements of the DGR in these genomes was confirmed using myDGR software. Ten (9%) investigated genomes had no elements of a DGR system in the genome region (Figure 5). The remaining 66 (73%) genomes demonstrated incomplete DGR-cassettes possessing truncated genes encoding T-CFP and RT with either the presence or absence of VR and TR (Figure 5, Appendix A). Eighty of the analyzed genomes contained VRs; however, only 30 of these genomes contained TRs. Despite the substantial variability of the genomic region containing DGR elements, ORFs encoding the tail needle protein, hypothetical protein, and integration host factor subunit were conserved between the investigated genomes, if the ORFs were present (Figure 5).

### 3.4. Analysis of VRs

Complete target genes encoding T-CFP were obtained from a number of genomes, including the crAssphage LMMB and reference p-crAssphage (NC_024711.1) (Appendix A), and the corresponding aa sequences were aligned. Then, similar VRs were identified in 57 complete T-CFPs (Figure 5, Appendix A). Additionally, the nucleotide sequences of TRs were extracted from 30 genomes and the appropriate aa sequences were aligned. This analysis revealed that the flanking regions of TRs aligned well with the same regions of VRs (Appendix A).

The consensus VR structure of crAssphages from *genus I* was calculated using WebLogo software and compared with the consensus structures of VRs from DGRs with RT of lineage 3 and VRs from CLec1, CLec2, and CLec3 classes (Figure 6). The calculated consensus VR structure was more conservative than the other structures, as it represented only sequences from *genus I* of the subfamily AlphacrAssvirinae, while the other consensus sequences comprised sequences from different members of large taxonomic groups such as Actinobacteria, Bacteroidetes, Firmicutes, and various prophages. It was revealed that the consensus VR structure of crAss-like phages of *genus I* had a similar motif to VRs from DGRs of lineage 3 and CLec3 VRs. The similarity was higher when comparing the consensus VR to VRs from DGRs of lineage 3; however, this similarity was still lower than the similarity between VRs from DGRs of lineage 3 and VRs of the CLec3 class (Figure 6).

## 4. Discussion

In this study, we describe the first finding of a DGR-cassette in the genome of the crAss-like phage. Gene synteny, phylogeny of the major capsid protein, and the absence of tRNA genes reliably confirmed that the studied crAssphage LMMB is a member of the candidate *genus I* of the subfamily AlphacrAssvirinae. The DGR-cassette containing all the essential elements (target gene-VR(IHF)-TR(IHF*)-RT) was found at the 3′-end of a cluster of structural genes, between the ORF51, which encodes the tail needle protein, and ORF48, which encodes the hypothetical protein. Based upon the DGR architecture, the fact that its RT belongs to lineage 3, and the absence of the avd gene, it was shown that the DGR of the crAssphage LMMB was a member of lineage 3.

Importantly, the genome of the crAssphage LMMB is not a unique crAss-like genome containing the DGR. Probably, this genetic element has not been identified in other crAss-like phages as the genome of the prototype p-crAssphage [1], which is usually used for annotation of genomes of the phages, does not contain the DGR. In fact, 14 other genomes from *genus I* of the subfamily AlphacrAssvirinae had orthologous DGRs with all essential elements. Usually, VRs from this type of DGR correspond to CLec3 [9]; however, the VRs in studied DGRs had some peculiarities. Although the consensus structure of VRs generated from the genomes of *genus I* showed a consensus motif with the consensus structures of VRs from DGRs with RTs of lineage 3 and CLec3 class VRs, this similarity was not high (Figure 6). The IMH sequences of the examined crAss-like phages also differed from the IMH sequences of DGRs of lineage 3 and CLec3 VRs. Through manual analysis, we showed that VRs of the phages have lost four aa (SRLC/A) in the IMH region compared to the VRs of DGR lineage 3 and CLec3. Previously, it has been reported that each VR within the class contains several conserved aa in their central regions, which are involved in the formation of the protein’s structural scaffold, and at their N- and C-ends, and these conserved aa are not subjected to A-to-N mutagenesis [9,27]. When we examined the VR sequences in the crAssphage LMMB and related phages to them, each adenine was found to undergo A-to-N mutagenesis (Appendix A). This leads to a higher theoretical level of VR mutagenesis in the target gene of crAss-like phages from *genus I* than in BPP-1 (10^16 aa vs. 10^13 aa). This level of mutagenesis is one potential reason why crAss-like phages are known as the most abundant viruses in the human gut.

However, our analysis showed that most of the genomes of crAss-like phages of *genus I* did not contain the DGR-cassette with all the essential elements. This fact indicates that, on the one hand, the DGR is not vital for the existence of crAss-like phages, and damage to the DGR does not lead to the elimination of such phages co-existing with a certain host in the relatively constant conditions of a healthy human gut. On the other hand, the instability of this region may be due to imperfect recombination occurring during mutagenic retrohoming.

In conclusion, the first DGR-cassette was discovered in the genomes of the crAssphage LMMB and several relative crAss-like phages. The DGRs had all the essential elements for retrohoming. However, complete DGR-cassettes were not found in most of the examined genomes of crAss-like phages. Therefore, we cannot conclude whether the DGR-cassette is an evolutionary advantage of crAss-like phages, or these DGRs were simply taken from the host genome. The functionality of DGRs in the crAss-like phages can only be proven in further experimental studies.

## Figures and Tables

**Figure 1 viruses-12-00573-f001:**
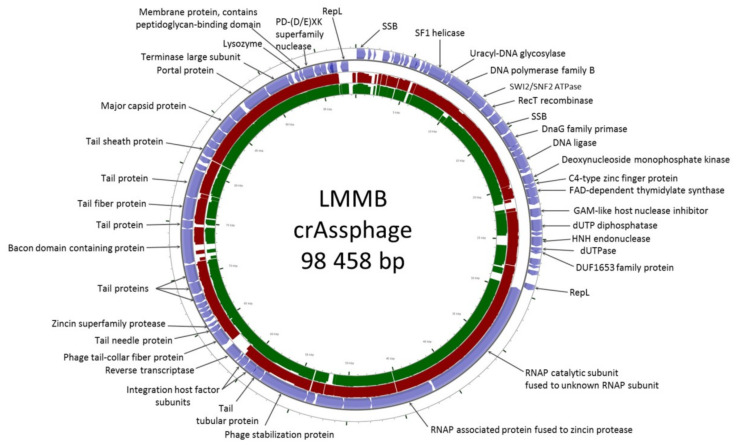
CrAssphage LMMB genome map visualized using the CGView server. Open reading frames (ORFs) are denoted with blue in the outer circle. The TBLASTX algorithm was used for sequence similarity comparison of the genomes of crAssphage LMMB, p-crAssphage [NC_024711.1] (similar genomic regions are marked with red), and NODE_105 [ODVQ01000105.1] (similar genomic regions are marked with dark green). Genomes of p-crAssphage and phage NODE_105 belong to *genus I* of the proposed subfamily AlphacrAssvirinae.

**Figure 2 viruses-12-00573-f002:**
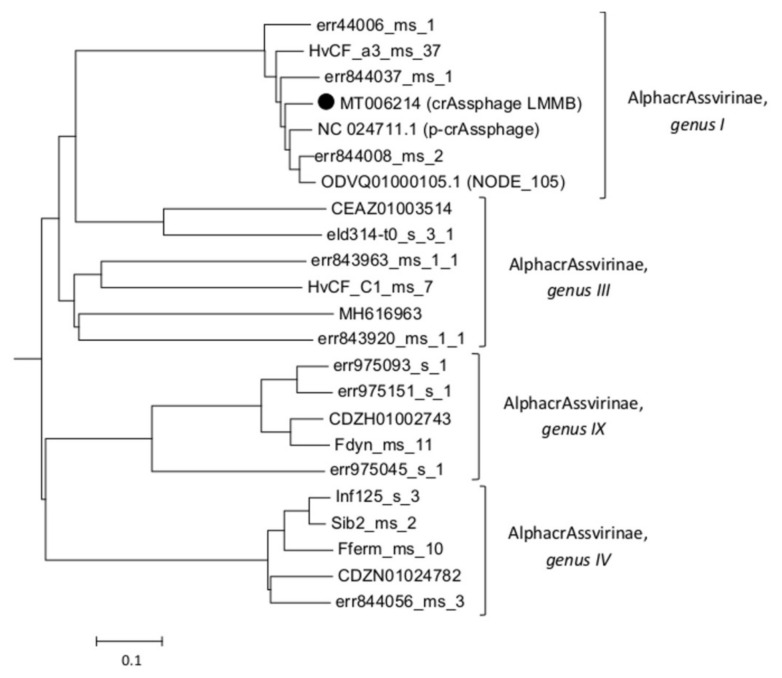
A phylogenetic analysis of the crAssphage LMMB and a number of crAss-like phages of the candidate subfamily AlphacrAssvirinae was performed using the Viral Proteomic Tree server. The investigated sequence is marked with a black circle. Genomes IDs are given. Genomes ODVQ01000105.1, CEAZ01003514, CDZH01002743, CDZN01024782, MH616963, and NC_024711 were downloaded from the GenBank databases, other sequences were extracted from the Supplementary data [3].

**Figure 3 viruses-12-00573-f003:**
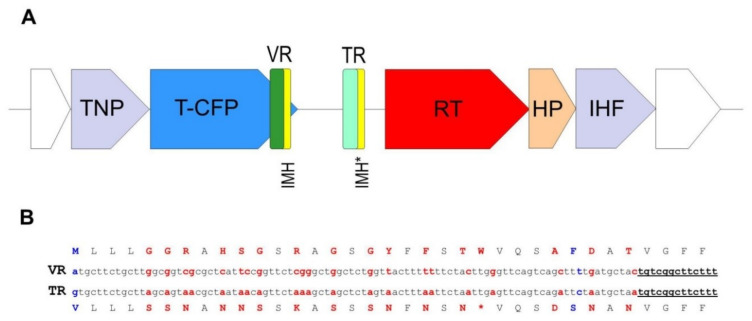
Schematic structure of the crAssphage LMMB diversity generating retroelement (DGR). (**A**) DGR in the genome of the crAssphage LMMB was verified using myDGR software (https://omics.informatics.indiana.edu/myDGR). The target gene (T-CFP) is marked with blue, the gene encoding reverse transcriptase (RT) is marked with red, the variable repeat (VR) is marked with green, the target repeat (TR) is marked with light green, both IMH and IMH* are marked with yellow. The flanking genes, encoding a tail-needle protein (TNP) and integration host factor (IHF) are marked with purple, the hypothetical protein (HP) is marked with orange. (**B**) Comparison of the VR and TR of the crAssphage LMMB shown for the DNA and aa sequences (The TR is not translated in vivo.). Variable positions in the VR and the corresponding adenine residues in the TR are marked with red letters. Non A-to-N mutations are indicated with blue letters. The IMH and IMH* sequences are underlined.

**Figure 4 viruses-12-00573-f004:**
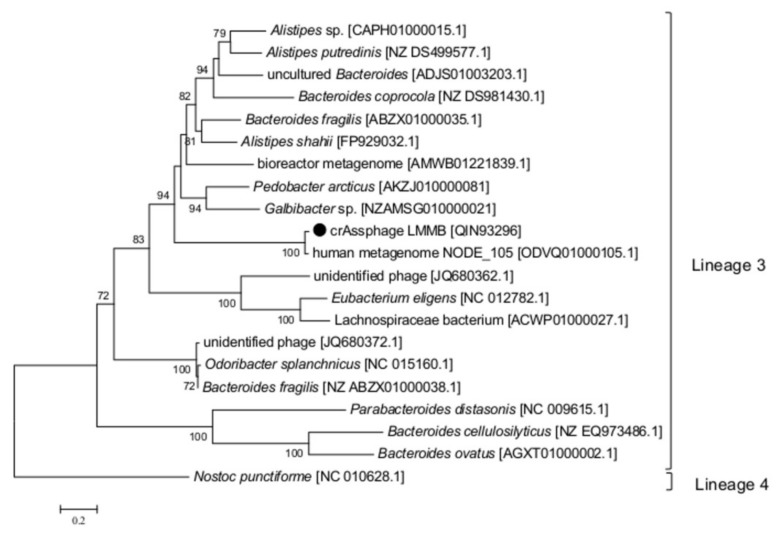
Phylogenetic analysis of the reverse transcriptase (RT) of the crAssphage LMMB and a number of similar RTs. The investigated sequence is marked with a black circle. The IDs of the genomes, which were used to extract gene sequences encoding RTs and translate them into aa sequences, are indicated in square brackets. Nucleotide sequences were extracted from the Supplementary data [9]. Phylogenetic analysis was performed using the maximum-likelihood method. The RT of *Nostoc punctiforme* (the Cyanobacteria phylum) was used as an out-group. Bootstrap values >70% are given at nodes.

**Figure 5 viruses-12-00573-f005:**
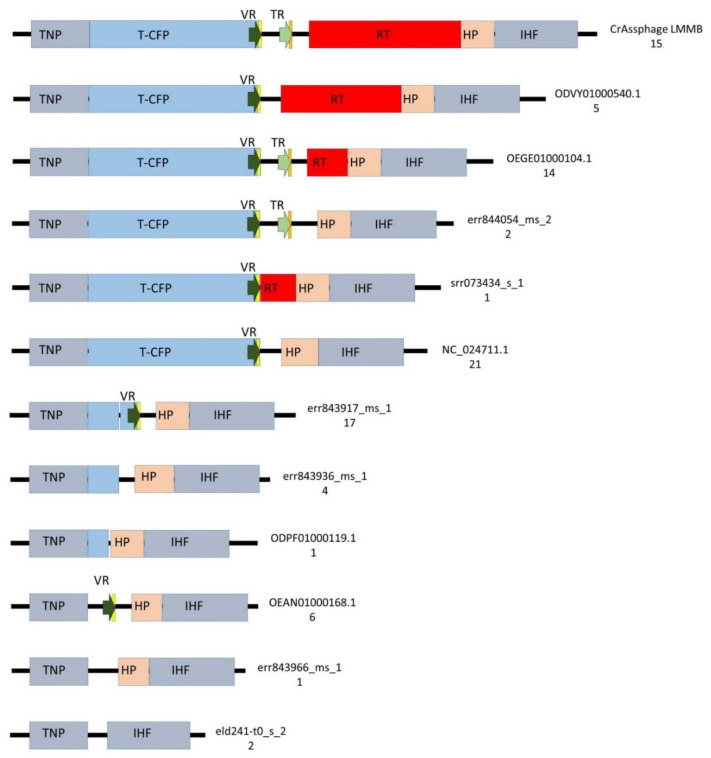
Schematic structure of the regions between the genes encoding tail needle protein (TNP) and integration host factor subunit (IHF) in 89 genomes of crAss-like phages from the candidate *genus I* of subfamily AlphacrAssvirinae. T-CFP — tail-collar fiber protein (marked with blue), RT—the gene encoding reverse transcriptase (marked with red), VR—variable repeat (marked with a dark green arrow), TR—template repeat (marked with a light green arrow), IMH—initiation of mutagenic homing sequence at the 3′-end of the VR (marked with yellow ), IMH*— a non-identical repeat of IMH at the 3′-end of TR (marked with orange), HP—the gene encoding the hypothetical protein (marked with beige), flanking genes IHF and TNP (marked with grey). ID number of the reference genome and a number of found genomes for each architecture are shown on the right side.

**Figure 6 viruses-12-00573-f006:**
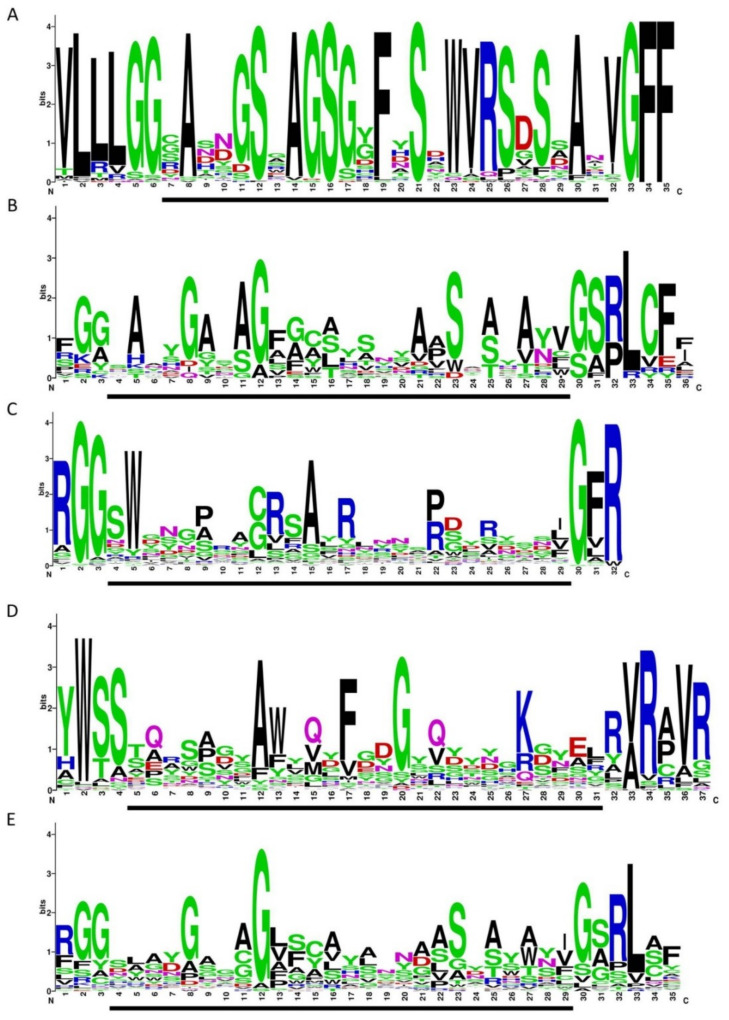
Variable repeat (VR) sequences shown in WebLogo format. (**A**) crAssphages from the candidate *genus I* of the subfamily AlphacrAssvirinae (extracted from 77 genomes). (**B**) VRs from diversity-generating retroelements (DGRs) with the reverse transcriptase (RT) of lineage 3. (**C**) CLec1 VRs. (**D**) CLec2 VRs (**E**) CLec3 VRs. Data used in the B–E were extracted from the Supplementary Material of Wu et al. [9] The regions involved in mutagenic retrohoming are indicated by a black bar under each WebLogo image.

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
