# Peer review of "First crAss-Like Phage Genome Encoding the Diversity-Generating Retroelement (DGR)"

_viruses, 2020, doi:10.3390/v12050573_

Round 1
Reviewer 1 Report
The work that is summarized in this manuscript points to the possibility that diversity-generating retroelements could be a source of variation in the AlphacrAssvirinae, extremely abundant phages in the human gut. As the authors have noted, the functionality and evolutionary significance will be investigated in future studies. The present work is very well presented and forms a very nice, complete story. Two very minor suggestions for improvement of the paper are offered.
- NODE_105 shows the greatest homology to crAssphage LMMB as indicated in Figure 1 and is one of the two phages that indicates that the newly discovered phage should be included in genus 1 of the AlphacrAssvirinae. As such, it was a bit surprising not to see NODE_105 included in Figures 2 and 4. Add this phage to these two figures?
- According to Figure 5, there were 14 phage sequences collected prior to crAssphage LMMB and that had complete DGRs. Perhaps a brief explanation as to why this element was not identified in this phage family prior to the present work could be included. Incomplete annotation of the earlier sequences?
Author Response
To the Reviewer 1:
We are sincerely grateful to the Reviewer 1 for comments that allowed us to improve our manuscript. We believe that this made the manuscript clearer and easier understood. Please, find below the item-by-item replies to your remarks.
The work that is summarized in this manuscript points to the possibility that diversity-generating retroelements could be a source of variation in the AlphacrAssvirinae, extremely abundant phages in the human gut. As the authors have noted, the functionality and evolutionary significance will be investigated in future studies. The present work is very well presented and forms a very nice, complete story. Two very minor suggestions for improvement of the paper are offered.
- NODE_105 shows the greatest homology to crAssphage LMMB as indicated in Figure 1 and is one of the two phages that indicates that the newly discovered phage should be included in genus 1 of the AlphacrAssvirinae. As such, it was a bit surprising not to see NODE_105 included in Figures 2 and 4. Add this phage to these two figures?
Yes, we add NODE_105 to Figures 2 and 4.
- According to Figure 5, there were 14 phage sequences collected prior to crAssphage LMMB and that had complete DGRs. Perhaps a brief explanation as to why this element was not identified in this phage family prior to the present work could be included. Incomplete annotation of the earlier sequences?
We believe that this genetic element has not been discovered because most of the crAss-like phage genomes are presented in the whole-genome shotgun (WGS) database. Only several genomes were annotated, however, those genomes do not contain complete DGRs or contain only incomplete DGR elements. As for the study [Guerin, 2018], the investigators used prototype p-crAssphage (NC_024711.1) for screening for previously unrecognized crAss-like phages from WGS for taxonomic investigation. However, the prototype p-crAssphage does not contain the DGR. We add this brief explanation in the Discussion. (lines 251-253).
Reviewer 2 Report
-
The authors have made an interesting observation, based on some minimal experimental work, but at the moment it is only that….an interesting observation. There is no functionality or biological context. To obtain this, they will need to undertake more experiments, particularly,
- identifying the host for this virus,
- then undertaking more extensive host range studies, as well as general phage characterisation
- such as one step growth curve,
- electron microscopy and
- general plaque morphology.
There are some methodological issues, as well as other issues, which are highlighted below. Specifically:
- Widely throughout manuscript: genus and species names aren’t italicised
- Widely throughout manuscript: use of both “phage” and “bacteriophage”
- Lines 75-77: more detail needed on the method of DNA extraction.
- Line 107: typical of, not typical to
- Line 117: so what would be the significance of no tRNAs?
- Lines 116-122 (methods for fig 2): looking solely at the major capsid protein is a very limited way to produce a phylogeny, especially when looking at viruses which the authors suggest are themselves diverse, and capable of generating diversity among prokaryotes. At the very least, whole genomes should be used, and newer techniques, such as VIPTree investigated to generate a more realistic picture of the functional variation between these viruses.
- Figure 6-for some of the smaller stacks it is not possible to decipher the identity of the aa at some positions….suggest that these are made clearer, or clarified elsewhere.
- Bacteroides intestinalis is suggested to be the host for the CrAss001 phage….I strongly suggest that the authors perform host range studies, using samples of the faecal material they used to isolate the DNA for their crAss-like phage, on a range of gut bacteria, in particular, Bacteroides This would help gain a fuller picture of the biology of the phage they have a sequence for. At the moment, it is just a sequence.
- At the end of the manuscript, the authors suggest that: “the functionality of DGRs in the crAss-like phages can only be proven in further experimental studies”….this is true, but the authors only have a sequence, and no biological data. A broader picture could be gleaned if the actual host, at least, could be identified, and then the important phage characterisation experiments performed, as outlined in a)-e) above.
Author Response
Please, see the attached file

Reviewer 3 Report
The manuscript reports on the presence of a diversity generating retroelement in a Bacteriodetes podovirus, and then gives a survey of the distribution of this element in related genomes. This is all tied up with metagenomics of an abundant but poorly sampled phage group and a novel host phage coadaption system. It is very interesting work, and I feel confident that the main results are accurate.
Major criticisms:
Although I know ICTV has gotten flaky about the designation "podovirus", please do mention that it's a podovirus. That's what establishes a background for most of the readers.
The GenBank submission has been blocked so that I can't see it. The labeling on the map reveals that it is poorly done. There is no such thing as a sheath protein in podoviruses (except for the atrociously misnamed N4 tail appendage). There are all sorts of "phage stabilization proteins"; this one is P22 gp10 phage stabilization protein. There are three different tail tubular proteins. This one is either P22 gp4 tail tubular protein, or T7 tail tubular protein A; I can't tell which. Those are the only two tail proteins on a podivirus. The others are tail appendages. This phage group has an over abundance of putative appendage proteins, so if there's some insight about what's going on here, please provide it. The genomic location of appendage genes relative to the core tail genes is often the only clue as to whether they are head, central tail or side fibers, so it's important to identify the core tail proteins correctly. It's unclear if the thing being called a "needle" is a homolog of P22 gp26 (which is a central tail fiber), or by what other criterion it would be called tail needle (or if it was meant to imply central fiber functioning in host recognition like P22 gp26). Be aware that appendage genes are mosaic, and be careful not to copy a name that belongs to a domain not actually matched by your gene. GenBank is full of head proteins that are named tail fibers because of this. Be sure to annotate homologs of phi14:2 mass spec. determined proteins, because we can at least be sure those are actually virion structural proteins. If there is some well characterized structural prototype closer than P22, annotate similarities to that also. Make a well annotated GenBank submission and show it to me, and I'll be satisfied. There is no need to make bunches of changes to the paper or pile stuff into supplementary data to meet this criticism.
Keep in mind that the BPP-1 DGR system modifies tail spikes (side fibers) not a tail needle (central fiber). Those different appendages play different roles in the phage infection process; so these aren't just semantic issues. The fuzziness of the annotation is messing with my ability to picture what your results might mean.
Minor criticisms:
Legend to fig. 2. "Genome IDs are given in square brackets." There are no square brackets on the figure.
line 151: "RT sequences extracted from the study of Wu et al. (2018)" Switch to numerical citation format.
line 255: "1016 aa vs. 1013 aa"; I think you mean 10^16 vs. 10^13 potential variant sequences.
Legend to fig. 5. IMH and IMH* are defined in the legend, but these acronyms do not appear on the figure. It should be "yellow box -- ... orange box --..." [or whatever color that IMH* box is]
Line 269: Therefore, we cannot conclude whether the DGR-cassette is an evolutionary advantage of
crAss-like phages, or these DGRs from the host genome. This sentence is garbled.
Author Response
Please, see the attachment.

Round 2
Reviewer 2 Report
I have looked at the authors’ responses, and some of the revisions in the paper. While the authors have addressed some of the issues I raised, they avoided what I thought was an important issue, that of identifying the host of this putative phage. They commented that this would take some time. I agree, but for completeness, I think it would be important. However, my area of research is that of the full functional and morphological characterisation of phages, for their potential to be considered for application in environmental remediation, or in therapy. The author’s focus is somewhat different to this, and their portrayal of the genomics of this relatively novel phage is of interest to those in their field.
Reviewer 3 Report
They've done what I asked. I have no further criticisms.